# Septin 7 interacts with Numb to preserve sarcomere structural organization and muscle contractile function

Rita De Gasperi[1,2], Laszlo Csernoch[3,4], Beatrix Dienes[3], Monika Gonczi[3], Jayanta K Chakrabarty[5], Shahar Goeta[5], Abdurrahman Aslan[2,6], Carlos A Toro[2,6], David Karasik[7], Lewis M Brown[5], Marco Brotto[8], Christopher P Cardozo[2,9]*

[1]Department of Psychiatry and Friedman Brain Institute, Icahn School of Medicine at Mount Sinai, New York, United States; [2]Spinal Cord Damage Research Center, James J. Peters VA Medical Center, Bronx, United States; [3]Department of Physiology, Faculty of Medicine, University of Debrecen, Debrecen, Hungary; [4]ELKH-DE Cell Physiology Research Group, University of Debrecen, Debrecen, Hungary; [5]Quantitative Proteomics and Metabolomics Center, Department of Biological Sciences, Columbia University, New York, United States; [6]Department of Medicine, Icahn School of Medicine at Mount Sinai, New York, United States; [7]Azrieli Faculty of Medicine, Bar Ilan University, Safed, Israel; [8]Bone-Muscle Research Center, College of Nursing & Health Innovation,University of Texas at Arlington, Austin, United States; [9]Department of Rehabilitation Medicine, Icahn School of Medicine at Mount Sinai, New York, United States

*For correspondence:
christopher.cardozo@mssm.edu

**Competing interest:** The authors declare that no competing interests exist.

**Abstract** Here, we investigated the mechanisms by which aging-related reductions of the levels of *Numb* in skeletal muscle fibers contribute to loss of muscle strength and power, two critical features of sarcopenia. Numb is an adaptor protein best known for its critical roles in development, including asymmetric cell division, cell-type specification, and termination of intracellular signaling. *Numb* expression is reduced in old humans and mice. We previously showed that, in mouse skeletal muscle fibers, Numb is localized to sarcomeres where it is concentrated near triads; conditional inactivation of *Numb* and a closely related protein *Numb*-like (*Numbl*) in mouse myofibers caused weakness, disorganization of sarcomeres, and smaller mitochondria with impaired function. Here, we found that a single knockout of *Numb* in myofibers causes reduction in tetanic force comparable to a double *Numb*, *Numbl* knockout. We found by proteomics analysis of protein complexes isolated from C2C12 myotubes by immunoprecipitation using antibodies against Numb that Septin 7 is a potential Numb-binding partner. Septin 7 is a member of the family of GTP-binding proteins that organize into filaments, sheets, and rings, and is considered part of the cytoskeleton. Immunofluorescence evaluation revealed a partial overlap of staining for Numb and Septin 7 in myofibers. Conditional, inducible knockouts of *Numb* led to disorganization of Septin 7 staining in myofibers. These findings indicate that Septin 7 is a Numb-binding partner and suggest that interactions between Numb and Septin 7 are critical for structural organization of the sarcomere and muscle contractile function.

## eLife assessment

This **convincing** study demonstrates a potentially **important** role for the factor Numb in skeletal muscle excitation–contraction coupling since a Numb knockout reduced contractile force. The authors thus demonstrate a novel role for Numb in calcium release in skeletal muscle.

**Figure 1.** Variants of *Numb*. (**A**) The PCR strategy used to detect *Numb* splice variants is shown. Red arrows depict the locations of the forward and reverse primers used. Exons (black boxes) are numbered 1–10. (**B**) cDNA was prepared using total RNA isolated from denervated (Dn) or control (**C**) gastrocnemius muscle and from undifferentiated (**U**) and differentiated (**D**) C2C12 myoblasts and amplified using primers spanning exons 1–6 (upper panel) or 8–10 (lower panel). PCR products were analyzed by agarose gel electrophoresis. Gastrocnemius muscles were removed at 7 d after left sciatic nerve transection. Differentiated C2C12 cells were harvested at 5 d after transferring cells to differentiation media. PTBL and PTBs refer to phosphotyrosine binding domain with or without exon 3; PRRs: proline-rich region, s refers to the PRR without exon 9, which is the only form present in muscle and myogenic cells.

## Introduction

A common consequence of aging is sarcopenia, which is characterized by declining skeletal muscle mass, muscle contractile force, muscle power, and overall physical function (***Brotto and Abreu, 2012***). The causes of reduced contractile function and power of skeletal muscle in individuals who have sarcopenia remain incompletely understood. One candidate mechanism is the reduced expression of the adaptor protein Numb in muscle of older organisms. *Numb* mRNA expression is reduced in muscle biopsy samples during normal aging in humans (***Carey et al., 2007***) and Numb protein levels are diminished in skeletal muscle of 24-month-old mice (***De Gasperi et al., 2022***). Numb is an adaptor protein that is highly conserved from *Drosophila* to humans. Numb contributes to asymmetric cell division, specification of cell fate, trafficking of cell surface proteins such as integrins, and turnover of signaling molecules such as Notch, Hedgehog, and p53 (***Yan, 2010***; ***Pece et al., 2011***; ***Reichardt and Knoblich, 2013***). A role in mitochondrial fission and fusion has also been suggested in certain contexts (***Liu et al., 2019***). In mammals, the *Numb* gene contains 10 exons and is expressed as one of four splice variants. The three shorter variants lack exons 3, 9, or both. Removal of exon 3 shortens the phosphotyrosine binding domain while removal of exon 9 truncates a proline-rich domain (***Figure 1***).

Within skeletal muscle, Numb has dual roles. Its expression in satellite cells is critical for their proliferation and for tissue repair after muscle injury (***George et al., 2013***). In skeletal muscle fibers, a conditional knockout (cKO) from myofibers of *Numb* and the closely related protein *Numb*-like (*Numbl*) reduced muscle contractility (***De Gasperi et al., 2022***). Examination of muscle from mice with a double knockout of *Numb* and *Numbl* by transmission electron microscope revealed perturbed muscle ultrastructure and altered mitochondrial morphology (***De Gasperi et al., 2022***). Ultrastructural changes included increased spacing of Z-lines, staircasing of Z-lines, altered mitochondrial morphology, and loss of the regular spacing of sarcoplasmic reticulum (***De Gasperi et al., 2022***). In vitro experiments using mouse primary myotube cultures found that knockdown (KD) of *Numb* reduced myoblast fusion and mitochondrial function, and delayed caffeine-induced calcium release (***De Gasperi et al., 2022***). A subsequent study found that knockout (KO) of *Numb* in the heart led to cardiac dilation and altered cardiac and skeletal muscle sarcomere structure (***Wang et al., 2022***).

Since Numbl protein levels are very low or undetectable in adult skeletal muscle, we have posited that the effects of KO of *Numb* and *Numbl* on skeletal muscle ultrastructure and function are likely attributable to loss of *Numb* expression (*De Gasperi et al., 2022*).

Molecular mechanisms by which *Numb* and *Numbl* organize the sarcomere and assure optimal calcium release during excitation–contraction coupling remain poorly understood. The interactome of Numb includes p53 (*Carter and Vousden, 2008*; *Colaluca et al., 2008*), Mdm2 (*Juven-Gershon et al., 1998*; *Yogosawa et al., 2003*), the LNX family of proteins which target Numb for degradation by the ubiquitin-proteasome pathway (*Dho et al., 1998*; *Rice et al., 2001*; *Nie et al., 2002*), and proteins involved in internalization and trafficking of membrane proteins, Eps 15 and α-Adaptin (*Santolini et al., 2000*). Co-immunoprecipitation (co-IP) experiments have shown that Numb binds sarcomeric α-actin and actinin (*Wang et al., 2022*) and Numb has been proposed to participate in sarcomere assembly (*Wang et al., 2022*). The possibility that Numb binds other proteins present in myofibers has not been tested.

The goal of the current study was to better understand the molecular mechanisms by which the cKO of *Numb* in skeletal muscle myofibers perturbed muscle weakness. We began by comparing effects of a single knockout (sKO) of *Numb* or a double knockout (dKO) of *Numb* and *Numbl* on ex vivo physiological properties of the *extensor digitorum longus* (EDL) muscle. We then examined Numb protein binding partners in C2C12 myotubes using a combination of immunoprecipitation (IP) and liquid chromatography coupled with mass spectrometry (LC/MS/MS) approaches. Our results identify Septins as Numb-binding partners and provide evidence that loss of Numb perturbs the organization of Septin 7 within myofibers.

## Results

### Effect of *Numb* and *Numb/Numbl* cKO on force generation

To begin, we compared properties of EDL muscle during ex vivo testing between mice with a single cKO of *Numb* or a double cKO of *Numb* and *Numbl*. As expected, Numb protein levels in muscle lysates were significantly lower in tamoxifen-treated HSA-MCM/*Numb*(f/f) mice and HSA-MCM/*Numb*(f/f)/*Numbl*(f/f) mice at 14 d after starting tamoxifen compared to vehicle-treated mice of the same genotype (*Figure 2*). Next, ex vivo contractile properties of the EDL muscle were compared for HSA-MCM/*Numb*(f/f) mice and HSA-MCM/Numb(f/f)/*Numbl*(f/f) mice at 14 d after starting induction with tamoxifen. A single cKO of Numb (HSA-MCM/*Numb*(f/f) mice) markedly reduced tetanic force and twitch force generation (*Figure 3A and B*). There was a genotype effect of single *Numb* cKO for fatigue index (*Figure 4A*) while time to peak tension and half relaxation time were not changed (*Figure 4B and C*). As expected, EDL muscle from mice with a double *Numb-Numbl* cKO (HSA-MCM/*Numb*(f/f)/*Numbl*(f/f)

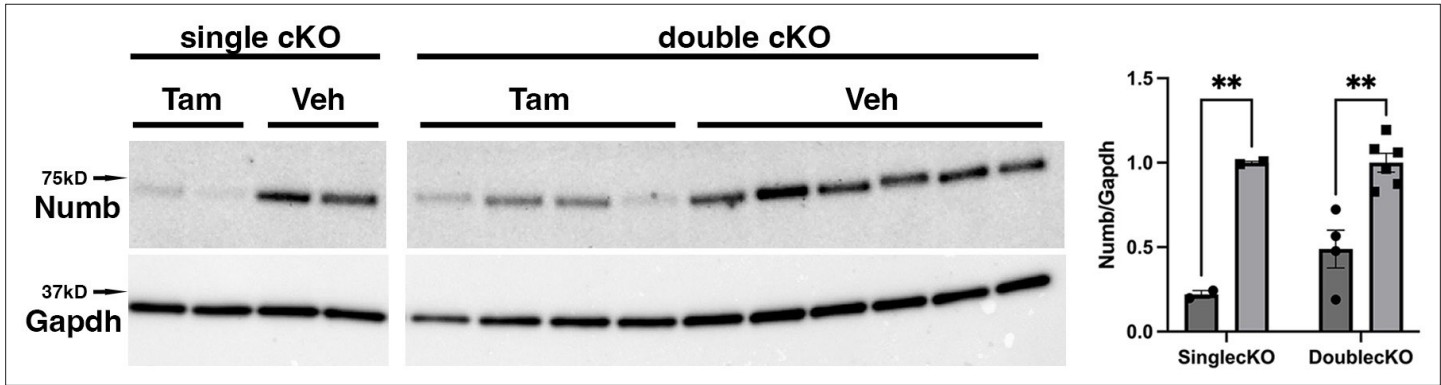

**Figure 2.** Representative Western blot showing downregulation of *Numb* levels in extensor digitorum longus (EDL) muscle from animals used for ex vivo physiology experiments. Data were analyzed by unpaired *t*-test. **p<0.01. Data as mean +/- STD. The lanes were cut from the same blot.

The online version of this article includes the following source data for figure 2:

**Source data 1.** Original western blot analysis of data shown *Figure 2* with anti-Numb (top image) and anti-GAPDH (bottom image).

**Source data 2.** *Figure 2* original western blot analysis (anti-Numb, top, and anti-GAPDH, bottom) with the bands shown in *Figure 2* highlighted and sample labels.

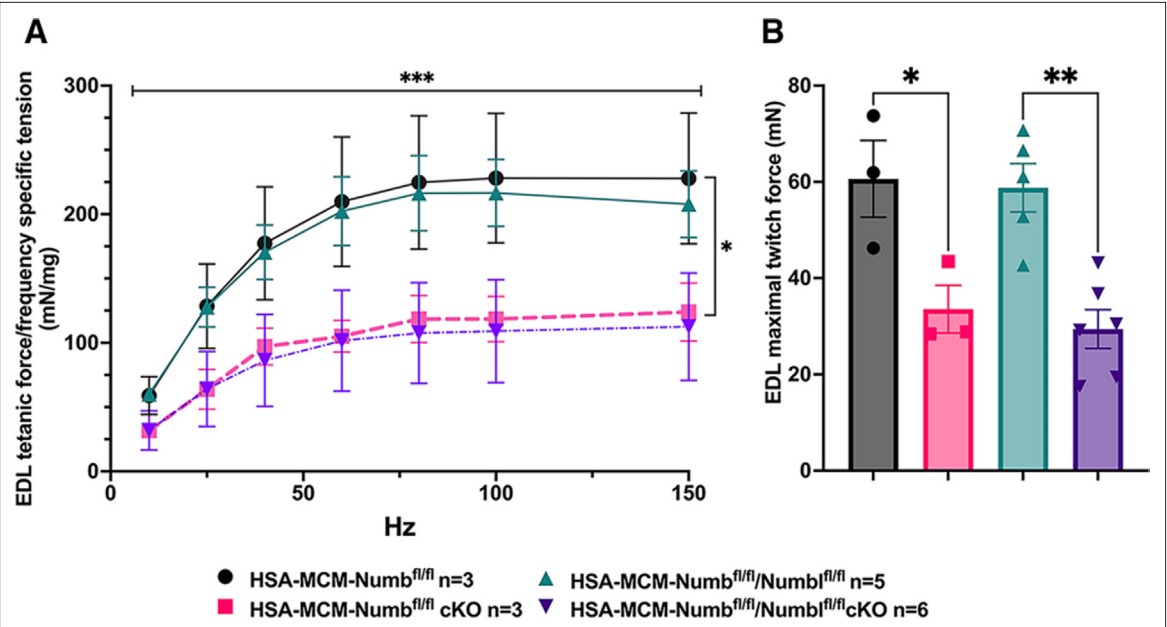

**Figure 3.** Effect of *Numb* or *Numb*/*Numbl* conditional knockout (cKO) on contractile function of extensor digitorum longus (EDL) muscle during ex vivo physiological testing. (**A**) Specific tension generated during tetanic contraction at the indicated frequencies is shown for weight normalized EDL muscle harvested from HSA-MCM/*Numb*(f/f) or HSA-MCM/*Numb*(f/f)/*Numbl*(f/f) mice at 14 d after starting injections of tamoxifen or vehicle. Statistical analysis was performed with repeated-measure ANOVA followed by Sidak's multiple-comparison test. $F$ = 28.29, DFn = 6, DFd = 22; ***p<0.001, force × frequency interaction ***p<0.001; (**B**) Maximum force generated during a single twitch is shown. Statistical analysis was performed with one-way ANOVA with Tukey's post hoc test ($F$ = 10.07, DFn = 3, DFd = 13). *p<0.05; **p<0.01. N = 3–6. Data expressed as mean +/- STD.

The online version of this article includes the following figure supplement(s) for figure 3:

**Figure supplement 1.** Force–frequency data from *Figure 3* were replotted for each group as percent maximal force produced for that group.

mice) demonstrated reductions in tetanic specific tension and maximum twitch force (*Figure 3A and B*) without any change in time to peak tension, half-relaxation time, or fatigue index (*Figure 4A–C*). There was no apparent difference for either specific tetanic tension or maximum twitch force when comparing single HSA-MCM/*Numb*(f/f) and double (HSA-MCM/*Numb*(f/f)/*Numbl*(f/f) mice) cKO mice.

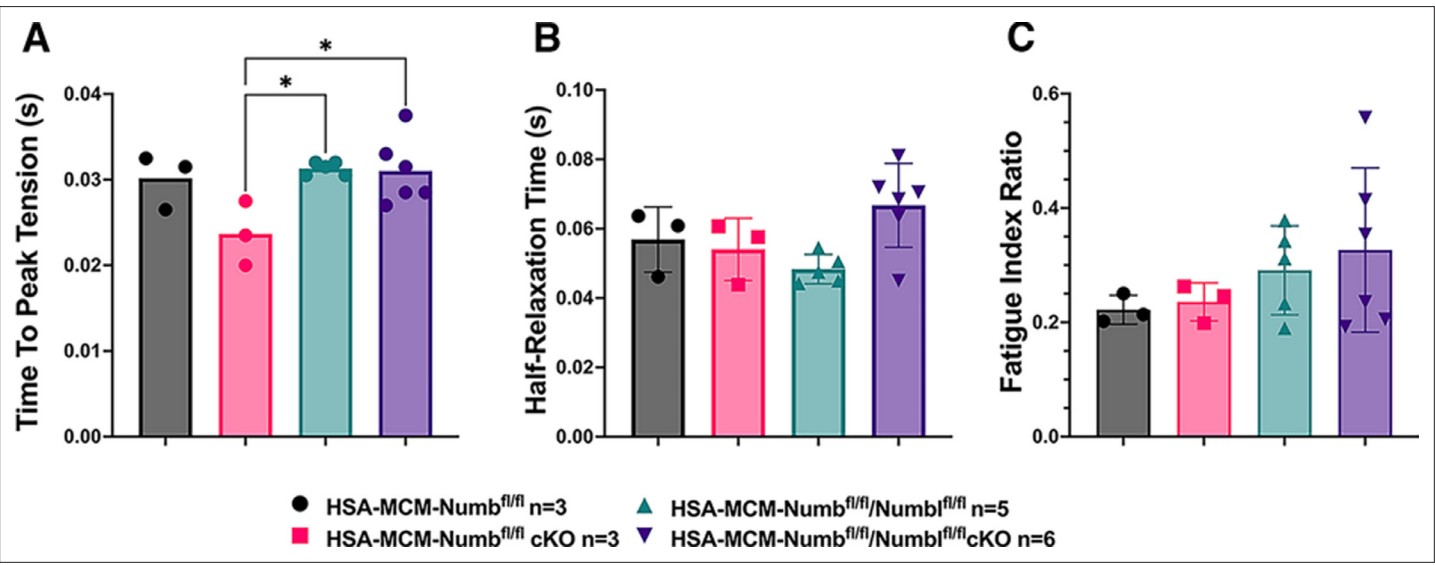

**Figure 4.** Time to peak tension (**A**), half-relaxation time (**B**), and fatigue index (**C**) are shown for HSA-MCM/*Numb*(f/f) and double (HSA-MCM/*Numb*(f/f)/*Numbl*(f/f) mice) at 14 d after starting injections of tamoxifen or vehicle. Data were analyzed by one-way ANOVA with Tukey's post hoc test. (**A**) $F$ = 4.561, DFn = 3, DFd = 13; *p<0.05; (**B**) $F$ = 3.679, DFn = 3, DFd = 13; (**C**) $F$ = 0.9772, DFn = 3, DFd = 13. N = 3–6. Datas expressed as mean +/- STD.

These data indicate that most, if not all, of the reduction of force-generating capacity observed in the *Numb/Numbl* double cKO line is attributable to inactivation of the *Numb* gene.

To understand if the relationships between frequency and tension were altered by *Numb* or *Numb/Numbl* cKO, data were replotted as the percent of maximum tetanic tension on the Y-axis and frequency on the X-axis. However, no shift in these curves was seen for either the *Numb* cKO or the *Numb* and *Numbl* dKO when compared to corresponding curves for vehicle-treated mice with normal expression levels of *Numb* and *Numbl* (*Figure 3—figure supplement 1*). These results confirm a critical role for *Numb* in force-generating capacity of skeletal muscle.

## Analysis of *Numb* Splice variants in the myogenic lineage

Given that *Numb* RNA can undergo alternative splicing to generate four different variants, and that alternative splicing shortens or removes protein–protein interaction domains, we sought to understand which splice variants of *Numb* were present in cells of the myogenic lineage in mice. Total RNA was extracted from mouse gastrocnemius muscle or from mouse C2C12 myoblasts then amplified using primers to sites flanking either exon 3 or exon 9 of mouse *Numb* mRNA (*Figure 1A*). No transcripts containing exon 9 were observed, and the majority of transcripts lacked exon 3, indicating that in skeletal muscle the primary form of *Numb* is that encoded by the shortest mRNA that lacks both exons 3 and 9. To gain some insight as to whether disease might alter *Numb* splice variants, this analysis was repeated for muscle at 7 d after sciatic nerve transection. No effect of denervation on splice variants present in muscle was observed (*Figure 1B*).

## Numb immunoprecipitation

Having confirmed that Numb is responsible for most, if not all, of the decrease in force production of muscle of our *Numb/Numbl* cKO mice, we sought to understand better how *Numb* participates in muscle force-generating capacity. We determined the binding partners for Numb in cells of the myogenic lineage by proteomics analysis of protein complexes isolated by immunoprecipitation using anti-Numb antibodies. For these experiments, we used C2C12 myotubes, which are multinucleated cell syncytia that express actin, myosin, and acetyl choline receptors and can be induced to contract by electrical stimulation. Using the methods described above, we were able to precipitate almost quantitatively Numb proteins when detergent was included in the cell lysis buffer. Yields were low when the detergent was removed, which we infer indicates that in C2C12 myotubes Numb is localized to membranes or is bound to membrane-bound proteins. Due to the incompatibility of LC/MS/MS analysis with most detergents commonly used in protein chemistry, choices of detergent that could be used were limited. After much optimization, we used a buffer containing a reduced amount of Triton-X100 (0.1%) for immunoprecipitation and washed the IP with TBS to remove as much Triton-X100 as possible. We also found that heating the samples at 37°C for 1 hr in 0.3% SDS was a mild yet effective elution method.

As shown in *Figure 5A*, using this method, we were able to immunoprecipitate Numb from 5-day differentiated C2C12 myotubes. Numb IP was verified by western blot using a rabbit monoclonal anti-Numb antibody whose specificity was previously validated by showing loss of Numb expression in C2C12 lysates treated with a specific vivo morpholino oligonucleotide (*De Gasperi et al., 2022*). Silver-stained SDS-PAGE gels revealed that immunoprecipitated proteins showed a uniform distribution of proteins across samples (*Figure 5B*).

## Mass spectrometry data analysis

LC/MS/MS analysis of peptides generated by digestion of the immunoprecipitated samples by trypsin followed by database searches identified 17,394 peptides. Mass spectrometry raw data files have been deposited in an international public repository (MassIVE proteomics repository at https://massive. ucsd.edu/) under data set # MSV000089327. The raw data files may be accessed by ftp protocol at ftp://massive.ucsd.edu/MSV000089327/. Using a false discovery rate (FDR) of 1% for both peptide sequence and protein identification, 1122 proteins were identified. Among these, 437 proteins were removed from analysis since they were represented by a single peptide or had insufficient data (<4 points /treatment) along with an additional 14 proteins that were derived from contaminants or added proteins. The analysis was conducted on 671 proteins that were represented by two or more peptides.

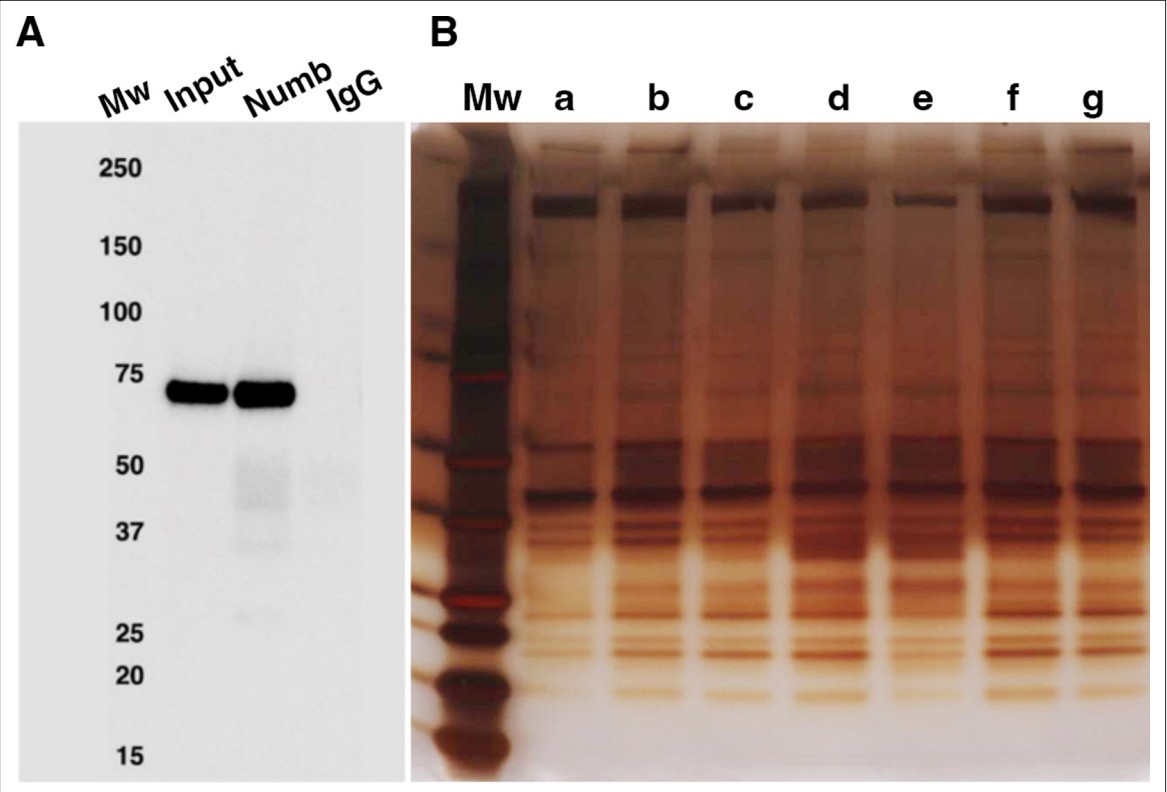

**Figure 5.** Numb immunoprecipitation. (**A**) Representative western blot of Numb immunoprecipitated from differentiated C2C12 myotubes. Input: original C2C12 lysate; Numb: goat anti-Numb IgG, control IgG: normal goat IgG. The blot was probed with a rabbit monoclonal anti-Numb antibody (**B**). Proteins recovered in each of the seven immunoprecipitations used for liquid chromatography coupled with mass spectrometry (LC/MS/MS) analysis were resolved by SDS-PAGE and visualized by silver staining.

The online version of this article includes the following source data for figure 5:

**Source data 1.** Original images of western blot analysis shown in *Figure 5A* (anti-Numb, top, and anti-GAPDH, bottom).

**Source data 2.** Original images of western blot analysis (anti-Numb, top, and anti-GAPDH, bottom) with the bands shown in *Figure 5A* highlighted and sample labels.

**Source data 3.** Original image of silver-stained IP gel shown in *Figure 5B*.

**Source data 4.** Original image of silver-stained IP gel with the bands shown in *Figure 5B* highlighted and sample labels.

To identify the proteins that may interact with Numb among the 671 that passed the first screen, the following criteria were used: p<0.01, two or more peptides identified, and detection of the protein target in four or more of the samples analyzed. Using these criteria, 11 potential Numb-binding proteins were identified (*Table 1*). A complete list of proteins for which peptide fragments were identified is given in *Supplementary file 1*.

## GWAS and WGS identify relationships of Numb-binding proteins and the skeleton

We took two approaches to identify potential links between these proteins and human physiology and disease. We first conducted searches of GWAS and WGS data (GWAS Catalog, https://www.ebi. ac.uk/gwas/home; MSK-KP, https://msk.hugeamp.org/; *Kiel et al., 2020*) for links between genetic variants in each of the genes encoding proteins identified as putative Numb-interacting proteins and diseases. Because of the extensive interactions of muscle and bone, our search included disorders of the skeleton and skeletal muscle. While no links to sarcopenia or other disorders of skeletal muscle were found, associations with several disorders of the skeleton were identified (*Supplementary file 2*) that included linkage between femoral neck bone mineral density (BMD) in women and phosphodiesterase 4D (PDE4D), and links between estimated BMD and Septin 9.

**Table 1.** Numb-interacting proteins identified by mass spectrometry.

| UniProt ID | Protein | Abundance ratio (Numb/control) | p-Value (control vs Numb) | Control count (out of 7) | Numb count (out of 7) |
|---|---|---|---|---|---|
| APCA4_MOUSE | Anaphase-promoting subunit 4 | ∞ | 0 | 0 | 7 |
| NIPA_MOUSE | Nuclear-interacting partner of Alk | ∞ | 0 | 0 | 7 |
| PDE4D_MOUSE | cAMP specific 3–5-cyclic phosphodiesterase 4D | ∞ | 0 | 0 | 6 |
| WNK1_MOUSE | Serine-Threonine kinase Wnk1 | 289 | $5.4E^{-4}$ | 4 | 7 |
| SEPT7_MOUSE | Septin 7 | 82.9 | $3.6E^{-3}$ | 4 | 7 |
| APC 10_MOUSE | Anaphase-promoting complex subunit 10 | ∞ | 0 | 0 | 5 |
| SEPT10_MOUSE | Septin 10 | ∞ | 0 | 0 | 4 |
| HOME3_MOUSE | Homer protein homologue | ∞ | 0 | 0 | 4 |
| TRAF7_MOUSE | E3 ubiquitin ligase Traf7 | ∞ | 0 | 0 | 4 |
| RUVB1_MOUSE | RuvB-like 1 | ∞ | 0 | 0 | 5 |
| SEPT2_MOUSE | Septin 2 | 379 | 0.01 | 5 | 7 |
| SEPT9_MOUSE | Septin 9 | 148 | 0.01 | 4 | 7 |

The online version of this article includes the following source data for table 1:

**Source data 1.** MS/MS data for Septin 7 from representative IP sample (67) showing peptides abundance and example mass spectra.

**Source data 2.** MS/MS data for cAMP-specific 3,5 cyclic phosphodiesterase 4D (PDE4D) from representative IP sample (520) showing peptides abundance and example mass spectra.

**Source data 3.** MS/MS data for nuclear -interacting protein of ALK (NIPA) from representative IP sample (1617) showing peptides abundance and example mass spectra.

**Source data 4.** MS/MS data for anaphase-promoting complex subunit 2 (ANC2) from representative IP sample (67) showing peptides abundance and example mass spectra.

**Source data 5.** MS/MS data for anaphase-promoting complex subunit 4 (APC4) from representative IP sample (1617) showing peptides abundance and example mass spectra.

**Source data 6.** MS/MS data for anaphase-promoting complex subunit 10 (APC10) from representative IP sample (520) showing peptides abundance and example mass spectra.

## Several Numb-binding partners are implicated in skeletal muscle function

A manually curated annotation was developed to understand the potential functional role of each of these proteins in skeletal muscle (*Supplementary file 2*). Sources used were GeneCards (*Safran et al., 2021*) and literature searches using PubMed. These curated data were then compared to the phenotype reported for myotubes and myofibers depleted of *Numb* and *Numbl* (*De Gasperi et al., 2022*), which includes reduced cell fusion, reduced mitochondrial function, delayed calcium transients, and muscle weakness. Homer3 bound Numb in our IP/LC/MS/MS assay and is similar to Homer1, a protein which has been implicated in excitation–contraction coupling (*Huang et al., 2007*). Septin 7 also bound Numb and was recently shown to cause multiple defects in skeletal muscle (*Gönczi et al., 2022*) that appear in many ways to phenocopy effects of KO of *Numb* and *Numbl* in myofibers (*De Gasperi et al., 2022*), including altered mitochondrial morphology and muscle weakness. In total, we identified four proteins belonging to the Septin family (Septin 2, 7, 9, and 10) as putative Numb-interacting proteins.

## Confirmation of Septin 7 interaction with Numb

Septins are GTP-binding proteins that are involved in many cellular processes by functioning as scaffolds to recruit other proteins or to compartmentalize cellular domains (*Gönczi et al., 2021*). Given the overlap in phenotype of skeletal muscle-restricted knockouts of *Numb* and *Septin 7* (*De Gasperi*

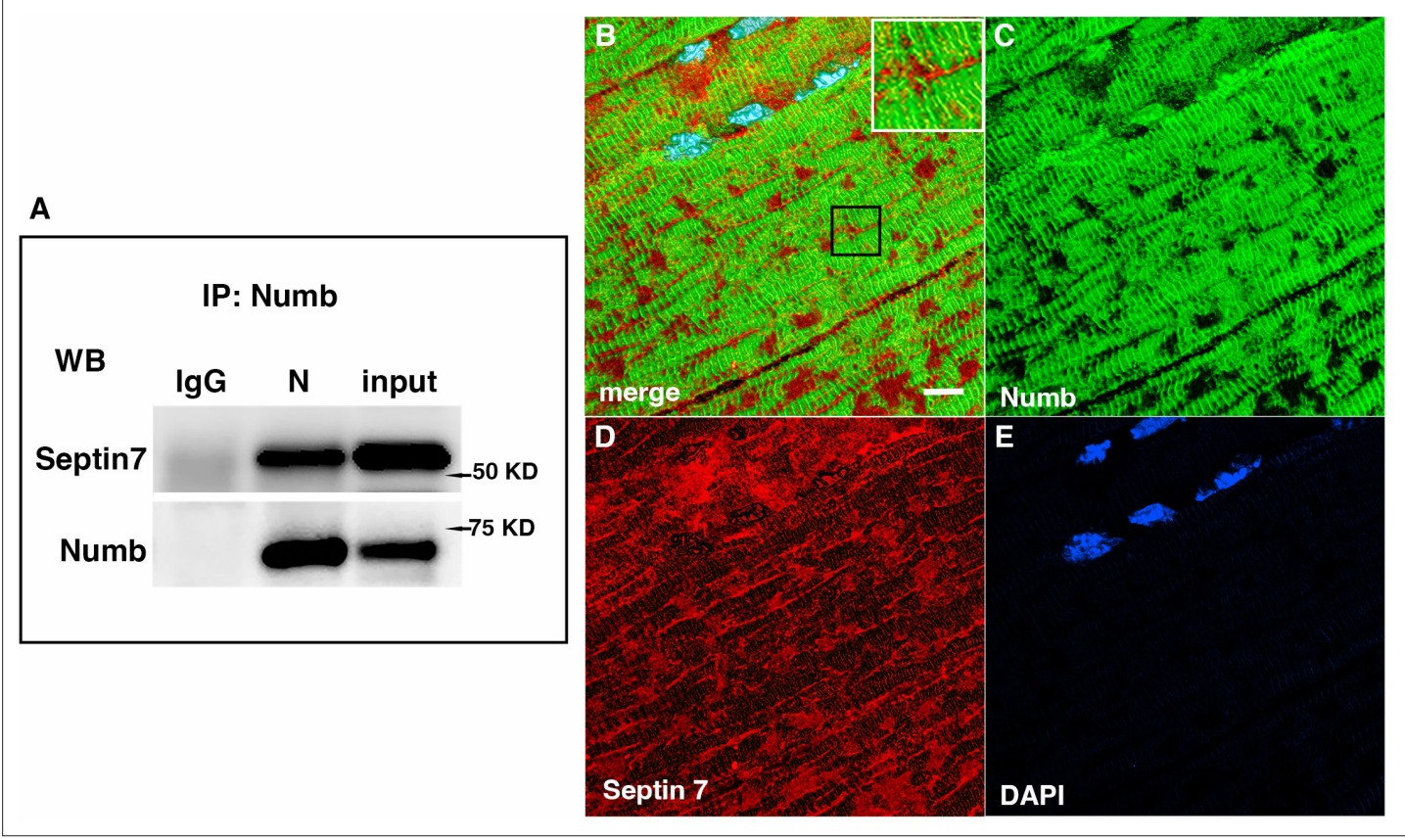

**Figure 6.** Western blot showing a representative co-immunoprecipitation of Septin 7 by anti-Numb antibody. (**A**) IgG: immunoprecipitation with control Ig; N: immunoprecipitation with anti-Numb; input: original C2C12 lysate. The upper panel was probed with anti-Septin 7, the lower panel with anti-Numb. (**B–D**) Immunolocalization of Numb and Septin 7 in muscle. Longitudinal cryosections of TA muscle from C57Bl6 mice were immunostained for Numb and Septin 7: (**B**) merged image; (**C**) anti-Numb antibody (green); (**D**) anti-Septin 7 antibody (red); (**E**) DAPI staining of nuclei (blue). Scale bar, 10 µm. The inset in (**A**) shows higher magnification (×2.3) of the area in the black square.

The online version of this article includes the following source data for figure 6:

**Source data 1.** Original images of western blot analysis shown in *Figure 6A* (anti-Septin, left, and anti-Numb, right).

**Source data 2.** *Figure 6A* images of western blot analysis (anti-Septin 7, left, and anti-Numb, right) with the bands shown in *Figure 6A* highlighted and sample labels.

*et al., 2022*; *Gönczi et al., 2022*), we sought to confirm their association biochemically and spatially by confocal microscopy. Using differentiated C2C12 myotubes grown independently from those used for the LC/MS/MS analysis, we performed co-immunoprecipitation followed by western blot and confirmed the binding of Septin 7 to Numb (*Figure 6A*). The pattern of immunostaining of Numb and Septin 7 was visualized by confocal microscopy (*Figure 6B–E*). In longitudinal sections of TA muscle, Numb immunostaining was seen as intense wavy bands partially traversing the myofiber. Septin 7 immunostaining was seen as intense streaks along the length of the myofiber with fainter bands traversing the fiber. *Figure 6B* shows the areas of Numb and Septin 7 co-localization.

Myonuclei demonstrate specialization of gene expression profiles based on location within the myofiber. To understand if regional variations in *Numb* or *Septin 7* expression might occur, we searched Myoatlas, a database of single-nucleus sequencing data (*Petrany et al., 2020*; https://research.cchmc.org/myoatlas/). *Numb* was expressed in myonuclei throughout the myofiber (*Figure 7A*) while, by contrast, *Septin 7* expression was most abundant in myonuclei expressing acetylcholine receptor subunits (*Figure 7B*). To understand if Septin 7 might be expressed at neuromuscular junctions, isolated myofibers were immunostained with anti-Septin 7 antibodies while acetylcholine receptors were labeled with α-bungarotoxin. Confocal microscopy imaging revealed greater intensity of Septin 7 immunolabeling at the neuromuscular junction (*Figure 7C*).

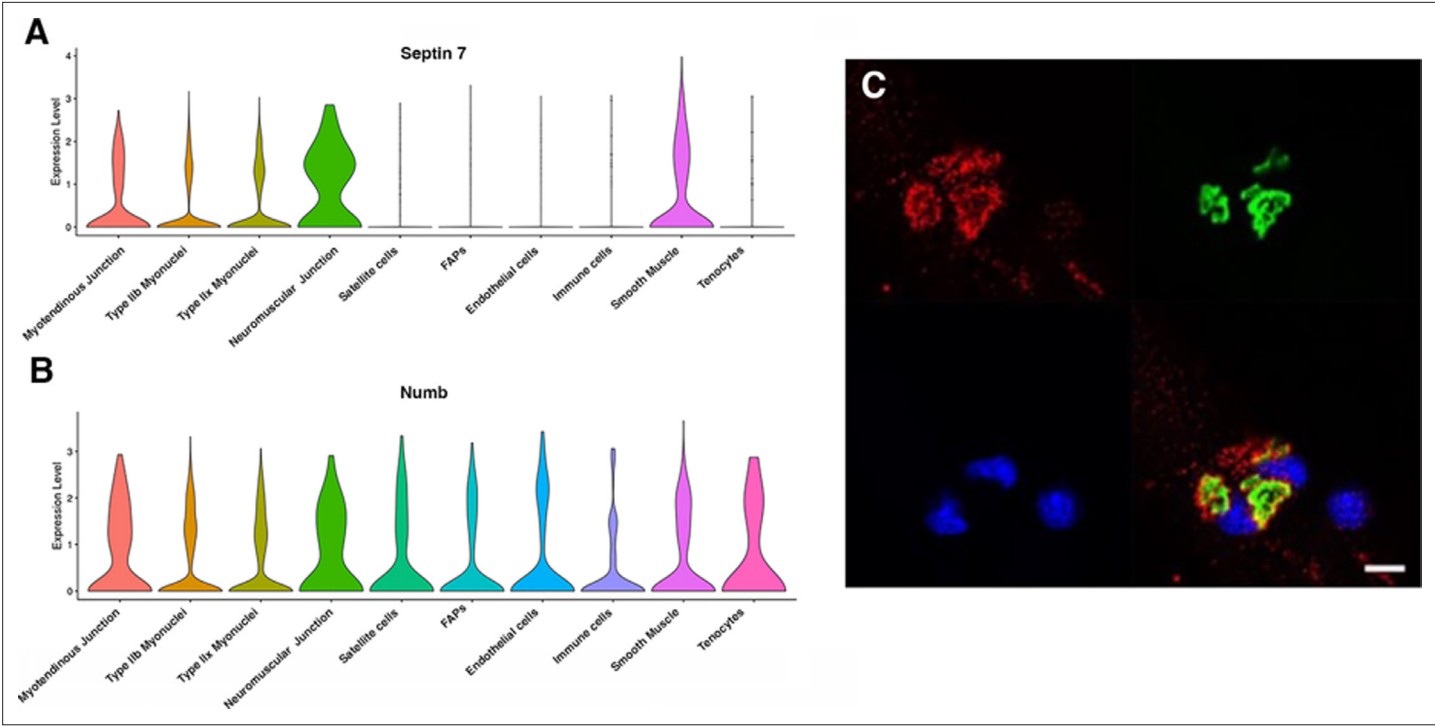

**Figure 7.** Septin 7 is enriched at neuromuscular junctions. (**A, B**) Violin plots generated by Myoatlas showing abundance of *Septin 7* (**A**) and *Numb* (**B**) mRNA in nuclei of TA muscle from 5-month-old mice (*Dho et al., 1999*). (**C**) Representative confocal microscopy image of the neuromuscular junction of an isolated myofiber immunostained with anti-Septin 7 (red) and with a fluorescently tagged α-bungarotoxin (green); nuclei were stained with DAPI (blue). Scale bar, 10 μm.

## Conditional knockout of *Numb* and *Numbl* perturbs Septin 7 organization

The marked perturbation of sarcomeric ultrastructure observed in mice with conditional, inducible knockdown of *Numb/Numbl* in skeletal muscle led us to ask whether the highly ordered localization of Septin 7 was also lost when Numb levels were reduced. Localization of Septin 7 was determined by immunostaining followed by confocal microscopy using single myofibers isolated from mouse hindlimb muscle from HSA-MCM/*Numb*(f/f)/*Numbl*(f/f) mice at 14 d after induction of *Numb/Numbl* knockout with tamoxifen (*Figure 8D–F*) or vehicle (*Figure 8A–C*). In *Numb/Numbl* cKO myofibers, the pattern of immunostaining of Septin 7 was more punctate while with loss of the ordered, linear staining along the axis of fibers and traversing them (*Figure 8D–F*).

## Discussion

The current study aimed to understand the molecular basis for the marked alterations in skeletal muscle ultrastructure, mitochondrial function, calcium release kinetics, and force production caused by cKO of *Numb* and *Numbl*. Findings that a single, conditional and inducible KO of *Numb* in skeletal muscle fibers resulted in marked reductions of peak twitch and peak tetanic force that were similar, if not identical, to those observed in a *Numb/Numbl* cKO confirm our conclusion that it was loss of *Numb*, rather than *Numbl*, that explained reduced muscle force production during in situ physiological testing in mice with a double *Numb/Numbl* KO (*De Gasperi et al., 2022*). The loss in force-generating capacity of EDL was observed by ex vivo physiological testing as soon as 14 d after the first injection of tamoxifen, indicating that deterioration of muscle in response to depletion of Numb occurs rapidly. These findings do not, formally, test if *Numbl* contributes to homeostasis of adult skeletal muscle fibers, although the very low-level expression of *Numbl* in adult skeletal muscle suggests that a role for this gene is unlikely (*De Gasperi et al., 2022*).

Our approach to understanding how *Numb* contributed to the molecular physiology of skeletal muscle contractility relied on LC/MS/MS analysis of tryptic peptides of proteins immunoprecipitated

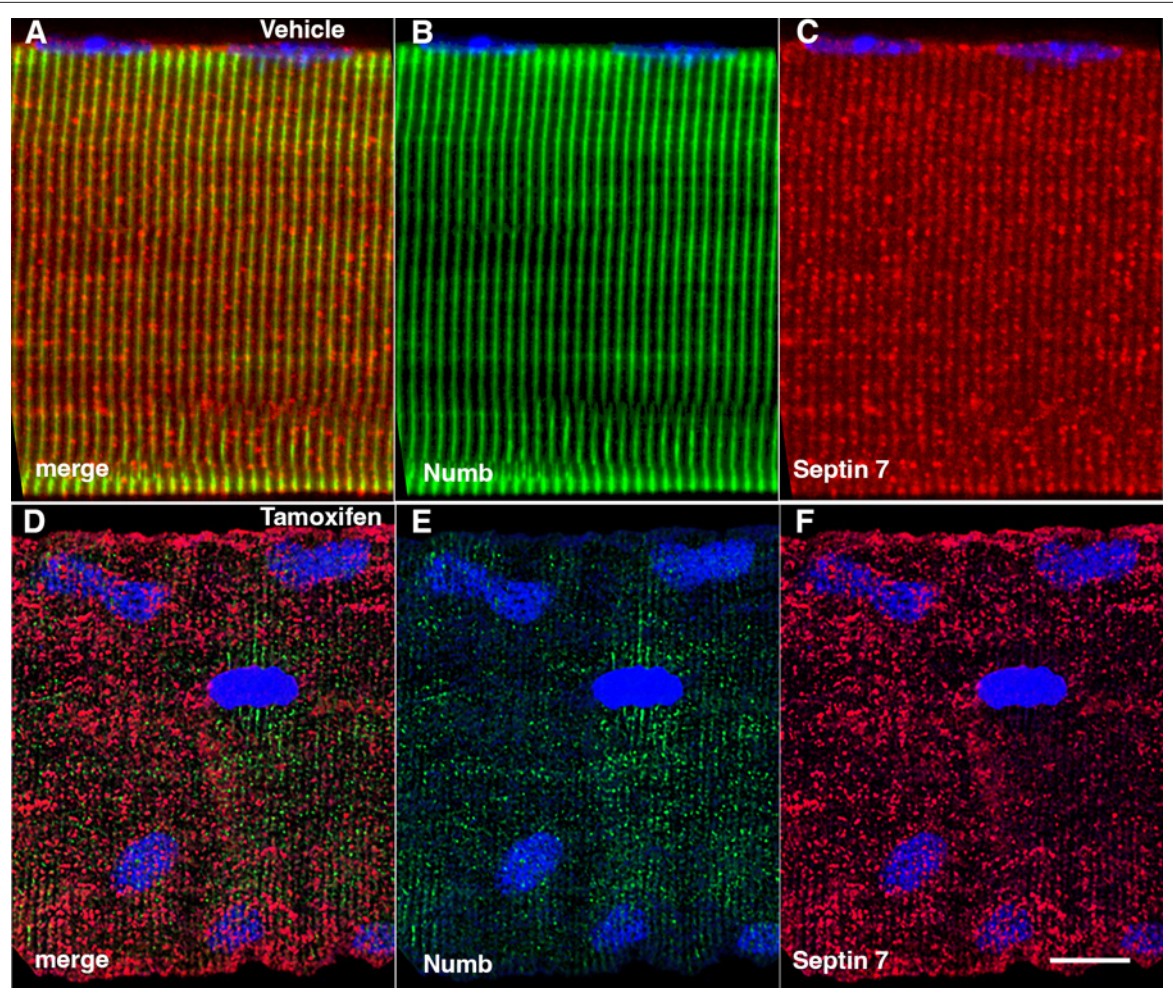

**Figure 8.** Representative confocal images of isolated mouse hindlimb fibers immunostained with anti-Numb (green) (**B, E**) and anti-Septin 7 antibodies (red) (**C–F**); merge: merged images (**A, D**). Nuclei were labeled with DAPI (blue). (**A–C**) Myofiber from a control mouse (vehicle treated); (**D–F**) myofiber from a Numb cKO mouse (tamoxifen treated). Scale bar, 10 µm.

from C2C12 myotubes using an anti-Numb antibody. By applying stringent criteria for identifying putative Numb-binding partners, a total of 11 high-probability Numb-binding proteins representing at least 6 different proteins or protein complexes was identified. None of these proteins were listed among the 209 known Numb-interacting proteins listed on the NCBI webpage for human Numb (https://www.ncbi.nlm.nih.gov/gene/8650#interactions). We believe these to be new, high-probability interactions, some of which may have muscle-specific roles.

The results support the conclusion that Septin 7 is an authentic binding partner of Numb within skeletal muscle fibers based on evidence that Numb pulls down Septin 7 from lysates of C2C12 myotubes, Numb and Septin 7 colocalize within skeletal muscle fibers, and cKO of *Numb* and *Numbl* perturbs the distribution of Septin 7 immunostaining. The identification of three other septins as Numb-binding partners is consistent with the findings that septins form hetero-oligomers that self-organize into fibrils that can polymerize into sheets, filaments, or rings (*Gönczi et al., 2021*). The co-localization of Numb with Septin 7 is constrained to specific regions of the myofiber suggesting overlapping but distinct functions of each protein in organization of the sarcomere. For example, Septin 7 was detected without Numb immunostaining in several locations, including around the nuclear envelope, and in longitudinal streaks that traverse several sarcomeres. Why these proteins interact at some locations but not others is unclear. Possibilities are that splice variants of Septin 7 vary in their distribution based on proteins they interact with and that only some splice variants bind Numb. It may also be that the interaction of Numb and Septin 7 is through a third yet to be identified protein that is localized

near the triad, our proposed localization for Numb (*De Gasperi et al., 2022*). Further study is needed to determine the explanation for the distribution of these two critical proteins.

Our findings suggest that Numb may also interact with other septins such as Septins 2, 9, and 10, which were also identified with a high level of confidence as Numb-interacting proteins by our LC/MS/MS analysis. Our data do not allow us to determine if Numb binds directly to these septins. Septins contain highly conserved regions, and, consequently, if one such region of Septin 7 interacts with Numb, then many septins would be expected to directly bind Numb through the same domain. However, because septins self-oligomerize, it is possible that when Numb binds to one septin, antibodies against Numb could also pull down other septins present in the septin oligomer to which Numb is bound regardless of whether or not they are also bound by Numb.

The roles in muscle physiology of Homer3, RUVBL1, Wnk1, and TRAF7 remain uncertain. Homer1, a protein closely related to Homer3, has been implicated in interactions between the Cav1.2 subunit of L-type calcium channels and ryanodine receptors (*Huang et al., 2007*) and is expressed in skeletal muscle. Its expression levels also correlate with serum alkaline phosphatase levels (*Sinnott-Armstrong et al., 2021*; *Supplementary file 2*). TRAF7 is an E3 ubiquitin ligase regulated by MyoD that ubiquitinates NF-kB targeting it for proteasomal degradation (*Tsikitis et al., 2010*) to facilitate myogenic differentiation. Zc3hc1 is another E3 ubiquitin ligase highly expressed in skeletal muscle known for its ability to target Cyclin B for proteolysis by the ubiquitin-proteasome pathway.

Our interest in the role(s) of Numb in skeletal myofibers was stimulated in part by the findings that its expression at the level of mRNA was reduced in older individuals (*Carey et al., 2007*), a finding we confirmed at the protein level in muscles from C57B6 mice (*De Gasperi et al., 2022*). The disorganization of Septin 7 immunostaining observed by day 14 after inducing *Numb* KD suggests that aging-related reduction of Numb expression may perturb organization of Septin 7 in a similar way though this prediction must be tested experimentally.

In conclusion, Numb binds Septin 7, a ubiquitously expressed protein involved in forming septin-based filaments, sheets, and cages that serve as the 'fourth component of the cytoskeleton'. The localization of these two proteins near the triad, together with binding of Homer 3 to Numb, strongly suggests that Numb and Septin 7 participate in organizing the triad through specific molecular interactions whose nature remains unclear. The interaction of Numb with septins has direct implications for understanding the molecular physiology of skeletal muscle and broader implications for understanding the roles of Numb and septins in biology.

## Materials and methods

### Animals

All animal studies were reviewed and approved by the James J. Peters Institutional Animal Care and Use Committee and were conducted in accordance with the requirements of the PHS Policy on Humane Care and Use of Laboratory Animals, the Guide, and all other applicable regulations. Animal protocol used CAR-18-29; IRBNet ID 1587044. C57BL/6NCrl mice were obtained from Charles River Laboratories (strain # 027). Transgenic mice in which *Numb* and *Numbl* can be conditionally ablated in skeletal muscle by injection of tamoxifen were previously described and are referred to as HSA-MCM/*Numb*^(f/f)/*Numbl*^(f/f) (*De Gasperi et al., 2022*). To generate mice in which a conditional, inducible KO of *Numb* in myofibers could be achieved, these mice were backcrossed with C57BL/6NCrl mice and bred until mice hemizygous for the HSA-MCM cassette and homozygous for the *floxed-Numb* allele were obtained. All mice were genotyped using genomic DNA isolated from ear snips as described (*De Gasperi et al., 2022*).

Knockout of *Numb/Numbl* or *Numb* was done by intraperitoneal injection of tamoxifen 2 mg/d for 5 d with one additional injection at day 10. The animals were sacrificed 14 d after the beginning of the induction. Controls were injected with vehicle (peanut oil). Animals were randomly assigned to the vehicle or tamoxifen group. Based on previous work (*De Gasperi et al., 2022*), we did not expect sex differences in ex vivo physiology so the data were combined across sex.

### Numb protein detection by western blot

Numb expression was evaluated by western blotting (*De Gasperi et al., 2022*; *Bubak et al., 2022*). *Tibialis anterior* (TA) and EDL were homogenized using an MP homogenizer in RIPA buffer (#9806, Cell

Signaling Technologies Inc, Danvers, MA) supplemented with a protease and phosphatase inhibitor cocktail (Halt, Thermo Fisher Scientific). The lysates were centrifuged at 14,000 rpm for 15 min and the supernatant saved. Protein concentration was determined with BCA reagent (Thermo Fisher Scientific). Then, 50 µg of protein were separated by SDS-PAGE and transferred to a polyvinylidene difluoride membrane using the Trans-Blot Turbo transfer pack (Bio-Rad Laboratories, Hercules, CA). Membranes were blocked for 1 hr at room temperature (RT) in Tris buffer saline 1% Tween-20 (TBS-T), 5% low-fat dry milk (blocking solution), and incubated overnight at 4°C with a rabbit monoclonal anti-Numb antibody (#2756, Cell Signaling Technology, Inc, Danvers, MA; RRID:AB_2154298) at 1:1000 dilution in 5% BSA in TBS-T. Membranes were washed in TBST and incubated with HRP-conjugated anti-rabbit IgG at 1:2000 dilution in blocking solution (#7074, Cell Signaling Technology; RRID:AB_2099233). Bands were revealed with ECL Prime reagent (RPN2236, Cytiva, Lifesciences, Piscataway, NJ) and imaged using the Amersham ImageQuant 800 imaging system (Cytiva). The blots were stripped and incubated with a rabbit monoclonal anti-GAPDH (1:1000 dilution) (#5174, Cell Signaling Technology; RRID: AB10622025) as loading control. Western blots were quantified using ImageQuant TL software (Cytiva).

## Tissue harvest and ex vivo physiology

Animals were weighed then anesthetized using inhaled 3% isoflurane. Hindlimb muscles were excised after careful blunt dissection. Measurement of whole-muscle contractile and mechanical properties was performed using an Aurora Scientific ex vivo physiology system for mice (Aurora, ON, Canada). A 4–0 silk suture was tied to the proximal and distal tendons of intact right EDL, immediately distal to the aponeuroses, the muscles were dissected and immediately placed in a bath containing a Krebs mammalian Ringer solution at pH 7.4, supplemented with tubocurarine chloride (0.03 mM) and glucose (11 mM) for 10 min. The bath was maintained at 25°C and bubbled constantly with a mixture of $O_2$ (95%) and $CO_2$ (5%). The distal tendon of the muscle was then tied to a dual-mode servomotor/force transducer (Aurora Scientific) and the proximal tendon tied to a fixed hook. Using wave pulses delivered from platinum electrodes connected to a high-power bi-phasic current stimulator (Aurora Scientific), each EDL was stimulated to contract. The 610A Dynamic Muscle Control v5.5 software (Aurora Scientific) was used to control pulse properties and servomotor activity, and record data from the force transducer. Optimal length (Lo) was established for each EDL to develop an isometric twitch force. EDL muscles were stimulated with a single electrical pulse to produce a twitch response. Stimulation produced a maximal twitch response by adjusting small increments (or decrements) to longer (or shorter) lengths. Muscle was left resting at least 45 s between twitch responses. Lo was achieved when twitch force was maximal. A frequency–force relationship was established once Lo was achieved. Here, EDL muscles were stimulated at increasing frequencies (i.e., 10, 25, 40, 60, 80, 100, and 150 Hz). Stimulation was delivered for 300 ms, and muscles were left to rest for 1 min between successive stimuli. Maximum absolute isometric tetanic force (Po) was determined from the plateau of the frequency–force relationship. Muscle fatigue resistance during repetitive stimulation at 60 Hz every second for a total of 100 stimuli (fatigue index [FI]) was also evaluated. Muscles were then removed from the bath solution and weighed. All data collected were analyzed using the Dynamic Muscle Analysis v5.3 software (Aurora Scientific).

## Cell culture

C2C12 cells (ATCC, ID CRL-1772; RRID:CVCL0188) validated and confirmed negative for mycoplasma by the supplier (ATCC) were grown in DMEM supplemented with 10% fetal bovine serum and antibiotics (Penicillin-Streptomycin [10,000 U/ml, Thermo Fisher]) until confluency was reached then switched to DMEM supplemented with 2% horse serum and antibiotics to induce differentiation and formation of myotubes. Cells were differentiated for 5 d, washed three times in phosphate-buffered saline solution (PBS), and harvested with a cell scraper. At harvesting time, the cells had fused to form myotubes. The pellets were kept at –80°C until used.

## Numb co-immunoprecipitation

Cell pellets were lysed in 25 mM Tris–HCl pH 7.2, 150 mM NaCl, 1 mM EDTA, 5% glycerol, 0.1% Triton X-100 supplemented with Halt protease and phosphatase inhibitor cocktail (Thermo Fisher) (IP buffer) for 30 min at 4°C with occasional mixing. The lysate was centrifuged at 14,000 rpm for 20 min and the

supernatant saved. Protein concentration was determined with the BCA reagent as per the manufacturer's instructions (ThermoFisher).

The extracts (1 mg protein) were pre-cleared with control 4%-Agarose resin (Thermo Fisher) for 1 hr at 4°C with constant rotation. The samples were centrifuged and the supernatant was immunoprecipitated with 1 µg of a goat polyclonal anti-Numb (Abcam, Ab4147; RRID:AB_304320), which has been previously used in IP (*Wu et al., 2010*; *Garcia-Heredia et al., 2017*) or with 1 µg of goat IgG (R&D Systems, Ab108-e) as IP specificity control for 1.5 hr at 4°C under constant rotation. To capture the complexes, 25 µl of Protein A/G Plus (Santa Cruz, sc-2003) were added and the samples incubated as above for 75 min. The samples were centrifuged and the beads washed twice with IP buffer, three times with 25 mM Tris–HCl, pH 7.2, 150 mM NaCl (TBS), and once with 4 mM HEPES in 10 mM HCl. Elution of the immunoprecipitated proteins was performed by heating the samples at 37°C for 1 hr with 0.3% SDS. The samples were centrifuged and supernatants were stored at –80°C.

Seven independent C2C12 samples were immunoprecipitated for the MS analysis, each pair (control IgG and anti-Numb) prepared on a different day from different cell culture samples. The Numb immunoprecipitated material was analyzed by SDS-PAGE and gels stained with silver stain kit (Bio-Rad).

## Verification of Numb immunoprecipitation

Small-scale immunoprecipitations were performed in parallel with the main IP to verify Numb IP. Proteins were separated by SDS-PAGE and blotted onto PVDF membranes. The membranes were blocked for 1 hr at 4°C in 50 mM Tris–HCl buffer, 0.15 M NaCl, 1% Tween-20 (TBS) supplemented with 0.5% non-fat dry milk (blocking solution) and incubated overnight at 4°C with a rabbit monoclonal anti-Numb antibody (Cell Signaling #2756; RRID:AB_2154298) diluted 1:1000 in TBS/5% BSA. After washing with TBS, the blot was incubated with HRP-conjugated anti-rabbit IgG (1:8000 dilution in blocking solution, Cytiva NA934; RRID:AB_772206), the blot developed with ECL Prime reagent (RPN2236, Cytiva) and imaged with an Amersham ImageQuant 800 imager (Cytiva).

## Preparation of immunoprecipitated samples for LC/MS/MS analysis

The volume of the immunoprecipitated samples was adjusted to 400 µl with MS grade water and 400 µl of methanol and 100 µl of chloroform were added. The samples were mixed with a vortex mixer for 1 min and centrifuged at 14,000 rpm for 1 min. The upper phase was removed without disturbing the proteins at the interphase and 400 µl of methanol was added. The samples were centrifuged at 14,000 rpm for 5 min to pellet the proteins. The pellet was washed three times with ice-cold methanol, dried for 10 min at RT, resuspended in 30 µl of freshly made 100 mM ammonium bicarbonate, 8 M urea, 0.1 M DTT, and flash-frozen in liquid nitrogen until analyzed. Cysteines were reduced and alkylated, and samples were loaded onto an S-trap micro column (Profiti C02-micro-80) according to the manufacturer's recommendations. Proteins were digested in the trap with trypsin, eluted, and lyophilized.

## LC/MS/MS analysis

The analysis was performed using Q Exactive HF (Orbitrap) mass spectrometer coupled to an UltiMate 3000 ultra-high-performance liquid chromatography (Thermo Fisher). The peptides were separated with reversed-phase chromatography using a 75 µm ID × 50 cm Acclaim PepMap reversed phase C18, 2 µm particle size column, and eluted from the Nano column with a multi-step acetonitrile/formic acid gradient. The flow rate was 300 nL/min.

For each sample, a 2.7 hr LC/MS/MS chromatogram was recorded in the data-dependent acquisition mode. The 15 precursor ions with the most intense signal in a full MS scan were consecutively isolated and fragmented to acquire their corresponding MS/MS scans. The full MS and MS/MS scans were performed at a resolution of 120,000 and 15,000, respectively. S-Lens RF was set at 55%, while the Nano-ESI source voltage was set at 2.2 kV.

The data were analyzed with MaxQuant_1.6.17.0. Peptide and fragments masses were searched against a database using the Andromeda search engine and scored using a probability-based approach that included a target decoy FDR. Data were then analyzed with Perseus 1.6.10.50. Searches were conducted against the UniProtKB Release 2019_07 (July 31, 2019) (*Mus musculus* sequences,

reviewed database with isoforms: 25,316 sequences) and included horse serum sequences, trypsin, keratins, and common lab contaminants.

## Septin 7 co-immunoprecipitation

To obtain independent confirmation of Numb association with Septin 7, Numb IP was performed with anti-Numb antibody as described above. For this analysis, we used three independently obtained samples of C2C12 myotubes different from those used for LC/MS/MS. The immunoprecipitated material was analyzed by western blot using anti-rabbit polyclonal anti-Septin 7 (Proteintech, 13818-1-AP; RRID:AB_2254298, 1:1000 dilution). To confirm Numb IP, the blot was then probed with anti-Numb antibody (Cell Signaling #2756). Secondary detection was performed with Clean-Blot IP detection reagent (Thermo Fisher, 21230, RRID:AB_2864363, 1:1500 dilution).

## Analysis of splice variants

At least four major *Numb* variant forms resulting from the alternative splicing of exon 3 and/or 9 have been found (*Dho et al., 1999*). To analyze *Numb* gene splice variants in muscle, total RNA was isolated from control and denervated gastrocnemius and from both undifferentiated and differentiated C2C12 using the Trizol reagent (Thermo Fisher), further purified by the RNeasy kit (QIAGEN) and reverse transcribed with the High Capacity cDNA reverse transcription reagents (Life Technologies). The cDNA was amplified by PCR using primers 5′TTCCCCCGTGTCTTTGACAG and 5′GTACCTCGGCCACGTAGAAG that span exons 1–6 to analyze exon 3 splicing and primers 5′ CTTGTGTTCCCAGATCACCAG and 5′ CCGCACACTCTTTGACACTTC spanning exons 8–10 (*Corallini et al., 2006*) to analyze exon 9 splicing. PCR was performed using 1 µl of cDNA and Top Taq DNA polymerase and buffer (QIAGEN). The reactions were performed for 30 s at 94°C, 45 s at 57°C, and 50 s at 72°C for 30 cycles. Aliquots of the PCR products were analyzed by 2% agarose gel electrophoresis. PCR products were cloned using the TOPO TA cloning system (Thermo Fisher) and multiple resulting clones were sequenced to confirm that the expected products were generated.

## Immunohistochemical staining

TA muscle from C57Bl/6 mice was snap-frozen in isopentane pre-cooled in liquid nitrogen. Longitudinal sections were cut using a cryostat. Sections were fixed for 7 min in cold 4% paraformaldehyde in PBS, washed five times with PBS, and blocked for 1 hr at RT in TBST/0.3% Triton X-100, 5% normal goat serum (blocking buffer). The sections were then incubated overnight with rabbit polyclonal anti-Numb (1:150 dilution, Cell Signaling #2756) and a rat monoclonal anti-Septin 7 (1:100 dilution, clone 19A4, MABT 1557, MilliporeSigma, Burlington, MA) in blocking buffer. Sections were washed with PBS and incubated with Alexa488-conjugated anti-rabbit IgG (A11008, Thermo Fisher) and Alexa568-conjugated anti-rat IgG (A11077, Thermo Fisher; RRID:AB_2534121) both at 1:300 dilution in blocking buffer for 2 hr at RT. The slides were washed in PBS, stained with DAPI (1 mg/ml in PBS), and mounted with Fluorogel mounting medium (EMS, Hatfield, PA). Immunostaining was visualized with a Zeiss LSM980 confocal microscope.

## Localization of Septin 7 in neuromuscular junctions

Enzymatically isolated single muscle fibers were fixed with 4% PFA for 20 min at RT. After the fixation, 0.1 M glycine in PBS was used to neutralize excess formaldehyde. Fibers were permeabilized with 0.5% Triton-X in PBS (PBST) for 10 min, blocked with a serum-free protein blocking solution (DAKO, Los Altos, CA) for 30 min, and rinsed three times with PBST solution. Anti-Septin 7 (JP18991, IBL, Hamburg, Germany) diluted in blocking solution was added and the fibers incubated overnight at 4°C in a humid chamber. Samples were washed three times with PBST and incubated with Alexa Fluor 488-conjugated alpha-bungarotoxin (1:500 dilution, B13422, Thermo Fisher) and Cy-3-conjugated anti-rabbit IgG at (dilution 1:300) (A10520, Thermo Fisher; RRID:AB_2534029) for 1 hr at RT. After washing three times, drop slides were mounted with DAPI containing mounting medium (H-1200-10, Vector Laboratories, Burlingame, CA). Images were acquired with a Zeiss AiryScan 880 laser scanning confocal microscope.

## Effect of *Numb/Numbl* cKO on Septin 7 protein expression

Hindlimb muscles from HSA-MCM/*Numb*[(f/f)]/*Numbl*[(f/f)] mice were dispersed by digestion with collagenase 1 as previously described (*De Gasperi et al., 2017*). Fibers were isolated at day 14 after inducing

*Numb/Numbl* KD with tamoxifen as described above. Individual fibers were briefly fixed as described previously (*De Gasperi et al., 2022*), incubated with anti-Septin 7 and anti-Numb antibodies as above, and visualized using Zeiss LSM 700 confocal microscope.

## GWAS and WGS searches

We conducted systematic searches of GWAS and WGS data (GWAS Catalog, https://www.ebi.ac.uk/gwas/home; Musculoskeletal Knowledge Portal, MSK-KP, https://msk.hugeamp.org/; *Kiel et al., 2020*).

## Statistics

Data are expressed as mean value ± standard deviation (STD). The significance of differences between groups was determined using either ANOVA or unpaired, two-tailed *t*-tests as described in the figure legends. Statistical calculations were performed with GraphPad Prism. A p-value <0.5 was used as the cutoff for significance.

## Acknowledgements

This work was supported by the U.S. Department of Veterans Affairs Rehabilitation Research and Development Service B9212C and B2020C to WAB, B7756R to CC, by NIH – National Institutes of Aging R01AG060341 (CC and MB) and PO1 AG039355 (MB), and the George W and Hazel M Jay professorship (MB). Also, Hungarian National Research, Development and Innovation Office funding scheme 2020-4.1.1-TKP2020, project no. TKP2020-NKA-04 (LC) and by the Ministry of Innovation and Technology of Hungary under the K_21 funding scheme project no. K_137600 (LC). The mass spectrometer was purchased under NYSTEM contract #C029159 from the New York State Stem Cell Science Board (LB) with matching funds from Columbia University and the Columbia Stem Cell Initiative.

## Additional information

### Funding

| Funder | Grant reference number | Author |
|---|---|---|
| U.S. Department of Veterans Affairs | I50RX002020 | Christopher P Cardozo |
| U.S. Department of Veterans Affairs | I01RX00776 | Christopher P Cardozo |
| National Institute on Aging | R-1AG060341 | Christopher P Cardozo |
| National Research, Development and Innovation Office | 2020-4.1.1-TKP2020 | Laszlo Csernoch |
| Ministry of Innovation and Technology of Hungary | K_137600 | Laszlo Csernoch |
| New York State Stem Cell Science | #C039159 | Lewis M Brown |
| Rehabilitation Research and Development Service | B7756R | Christopher P Cardozo |

The funders had no role in study design, data collection and interpretation, or the decision to submit the work for publication.

### Author contributions

Rita De Gasperi, Conceptualization, Formal analysis, Supervision, Funding acquisition, Investigation, Methodology, Writing – review and editing; Laszlo Csernoch, Conceptualization, Resources, Supervision, Writing – review and editing; Beatrix Dienes, Monika Gonczi, Formal analysis, Investigation, Methodology; Jayanta K Chakrabarty, Shahar Goeta, Abdurrahman Aslan, Investigation, Methodology;

Carlos A Toro, Formal analysis, Supervision, Investigation, Methodology, Writing – review and editing; David Karasik, Conceptualization, Resources, Supervision, Investigation, Writing – review and editing; Lewis M Brown, Conceptualization, Supervision, Funding acquisition, Investigation, Methodology, Project administration, Writing – review and editing; Marco Brotto, Conceptualization, Resources, Supervision, Funding acquisition, Project administration, Writing – review and editing; Christopher P Cardozo, Conceptualization, Funding acquisition, Writing – original draft, Project administration

## Author ORCIDs
Rita De Gasperi ⬤ http://orcid.org/0000-0002-5281-4263
Laszlo Csernoch ⬤ http://orcid.org/0000-0002-2446-1456
Abdurrahman Aslan ⬤ http://orcid.org/0000-0002-6730-4768
Carlos A Toro ⬤ http://orcid.org/0000-0002-7355-7534
David Karasik ⬤ http://orcid.org/0000-0002-8826-0530
Christopher P Cardozo ⬤ https://orcid.org/0000-0003-4013-2537

## Ethics

All animal studies were reviewed and approved by the James J. Peters Institutional Animal Care and Use Committee, and were conducted in accordance with requirements of the PHS Policy on Humane Care and Use of Laboratory Animals, the Guide and all other applicable regulations.Animal protocol used (CAR-18-29; IRBnet ID 1587044).

Reviewer #1 (Public review): https://doi.org/10.7554/eLife.89424.4.sa1
Reviewer #2 (Public review): https://doi.org/10.7554/eLife.89424.4.sa2
Author response https://doi.org/10.7554/eLife.89424.4.sa3

# Additional files

## Supplementary files

• Supplementary file 1. List of MS/MS data for all the proteins for which peptides fragments were identified in the IP samples.

• Supplementary file 2. Manually curated annotation for the Numb-binding proteins identified by IP-MS/MS showing domains, function, and associated diseases including linkage to disorders of the skeleton.

• MDAR checklist

## Data availability

Mass spectrometry raw data files have been deposited in an international public repository (MassIVE proteomics repository at https://massive.ucsd.edu/) under data set # MSV000089327. The raw data files may be accessed by ftp protocol at ftp://massive.ucsd.edu/MSV000089327/. Data is publicly available at:https://massive.ucsd.edu/ProteoSAFe/dataset.jsp?task=d445fc58f77e44008807df6f21fee036.

The following dataset was generated:

| Author(s) | Year | Dataset title | Dataset URL | Database and Identifier |
|---|---|---|---|---|
| De Gasperi R, Csernoch L, Dienes B, Gonczi M, Chakrabarty JK, Goeta S, Aslan A, Toro CA, Karasik D, Brown LM, Brotto M, Cardozo CP | 2023 | Septin 7 Interacts With Numb To Preserve Sarcomere Structural Organization And Muscle Contractile Function | https://massive.ucsd.edu/ProteoSAFe/dataset.jsp?task=d445fc58f77e44008807df6f21fee036 | MassIVE, MSV000089327 |

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
