## [Editor Report · eLife assessment]

This **convincing** study demonstrates a potentially **important** role for the factor Numb in skeletal muscle excitation–contraction coupling since a Numb knockout reduced contractile force. The authors thus demonstrate a novel role for Numb in calcium release in skeletal muscle.

---

## [Referee Report · Reviewer #1 (Public review)]

The authors investigate the function of the PTB domain containing adaptor protein Numb in skeletal muscle structure and function. In particular, the effects of reduced Numb expression in aging muscle is proposed as a mechanism for reduced contractile function associated with sarcopenia. Using ex-vivo analysis of conditional Numb and Numblike knockout muscle the authors demonstrate that loss of Numb but not the related Numblike gene expression perturbs muscle force generation. In order to explore the molecular mechanisms involved, Numb interacting proteins were identified in C2C12 cell cultured myotubes by immunoprecipitation and LC-MS/MS. The authors identify Septin 7 as well as Septin 2, 9 and 10 as a Numb binding proteins and demonstrate that loss of Numb/Numblike in myofibers causes changes in Septin 7 subcellular localization. Of note, whether additional septins form a complex or are also disrupted by Numb/Numblike loss remains an interesting area for further investigation. Additional investigation of the specificity and mapping of the Numb-Septin 7 (or another Septin) interaction would be of interest and provide an approach for future studies to demonstrate the biological relevance and specificity of the Numb-Septin 7 interaction in skeletal muscle

---

## [Referee Report · Reviewer #2 (Public review)]

Summary:

The main purpose of this investigation was to (1) compare the effects of a single knockout (sKO) of Numb or a double knockout (dKO) of Numb and NumbL on ex-vivo physiological properties of the extensor digitorium longus (EDL) muscle in C57BL/6NCrl mice; and (2) analyze protein complexes isolated from C2C12 myotubes via immunoprecipitation and LC/MS/MS for potential Numb binding partners. The main findings are (1) the muscles from sKO and dKO were significantly weaker with little difference between the sKO and dKO lines, indicating the reduced force is mainly due to the inactivation of the Numb gene; and (2) there were 11 potential Numb binding proteins that were identified and cytoskeletal specific proteins including Septin 7.

Strengths:

Straight-forward yet elegant design to help determine the important role the Numb has in skeletal muscle.

Weaknesses:

There were a limited number of samples (3-6) that were used for the physiological experiments; however, there was a very large effect size in terms of differences in muscle tension development between the induced KO models and the controls.

---

## [Author Response]

The following is the authors’ response to the previous reviews.

**Reviewer #1 (Recommendations for The Authors):**
1. While the specificity of the observed muscle phenotypes seems clear, the subsequent molecular analysis of Numb protein interactors does not seem to consider the potential involvement of Numb-like. The authors should demonstrate the relative expression levels of Numb and Numb-like in the models used, and establish the specificity of the antibodies used in IP, western and staining experiments.

Response: Perhaps the most convincing evidence that the anti-Numb antibody did not pull down Numb-like is that this protein was not detected among immunoprecipitated protein complexes pulled down by the anti-Numb antibody used. The antibody used in the immunoprecipitation was validated by the supplier and was previously reported to immunoprecipitate Numb [1, 2]. We previously demonstrated that a morpholino against Numb mRNA almost completely eliminated the band detected by this antibody and that this band was at the expected molecular weight [ref]. In our hands, mRNA levels for Numb-like in skeletal muscle are 5-10-fold lower than those for Numb [3]. We have been unable to detect Numb-like protein in healthy adult skeletal muscle by immunoblotting or immunofluorescence staining. Taking all of these findings together, it seems unlikely that the antibodies used for immunoprecipitating Numb-protein complexes pulls down Numb-like.

2. The authors use PCR to investigate Numb isoform expression and conclude that p65 is likely the dominant protein isoform expressed. While this agrees with the single band observed in Supp Figure 4A, a positive control for exon 9 excluded and included isoforms in the PCR reactions would strengthen this conclusion.

Response: The amplicons shown in Supplemental 4 were sequenced. The clones corresponded to the isoforms with the exon 3 present or removed. No amplicons containing exon 9 were detected. The following sentence was added to the Analysis of Splice Variants section of Methods to address this point: “PCR products were cloned using the TOPO TA cloning system (ThermoFisher) and multiple resulting clones were sequenced to confirm that the expected products were generated.”

3. PCR analysis of total Numb and Numb-like expression levels are not shown. This is important given the specificity of the Numb antibodies used for AP-MS experiments are not described and some Numb antibodies are well known to also recognize Numb-like. Two different Numb antibodies were used for Western and immunoprecipitation but the specificity for Numb and Numb-like is not described. In particular, does the antibody used in the AP-MS experiment recognize both Numb and Numb-like? Supplementary Table 1 does not list Numb or Numb-like, but presumably peptides were identified?

Response: As noted above, the specificity of anti-Numb antibodies was confirmed in previous studies [3]. Importantly, Numb-like mRNA levels are 5-10-fold lower than Numb mRNA, and NumbL protein is undetectable in healthy adult skeletal muscle by Western. The physiology data reported in this manuscript supports the conclusion that a single KO of Numb is sufficient to recapitulate the physiological phenotype of Numb/Numb-like KO . We therefore reason that the majority, if not all, of the physiological contribution of these proteins to muscle contractility due to Numb (Fig. 1).

4. The validation experiment used the same Numb antibody for immunoprecipitation, immunoblotted with Septin 7. A reciprocal IP of Septin 7 and blotted with Numb should be performed. In addition, a Numb-like IP or immunoblot would also be useful to demonstrate the specificity of the interaction. Efforts to map the interaction between Numb and Septin 7 would be useful to demonstrate specificity of the interaction and strategies to establish the biological relevance of the interaction.

Response: We agree with the reviewer and attempted several IPs with anti-Septin7 antibodies. These were unsuccessful. In a new collaboration, Dr. Italo Cavini (University of Sao Paulo) has used machine-learning-based approaches to model binding between Numb and several septins, including Septin 7. The analysis suggests that binding of Numb with septins involves a domain of Numb that has not yet been ascribed a function in protein-protein interactions. These computational predictions require experimental validation but provide rational starting point for experiments to define the domains responsible for these interactions. Such experiments were included in our recent NIH R01 renewal application. We hope to be able to report on results of confirmatory experiments of these computational models in the future.

5. Other septins were identified in the AP-MS experiment and might have been anticipated to also be disrupted by Numb/Numb-like deletion. Are these septins known to interact in a complex?

Response: This is an excellent question. Septins have conserved motifs providing a clear reason to imagine that many different mammalian septins could directly interact with Numb. Septins form heterooligomers consisting of complexes formed by 3, 6 or 8 septins [4]. It is likely that when Numb binds to one septin, antibodies against Numb pull down other septins present in the septin oligomer to which Numb is bound. The following paragraph was added to the discussion: “Our findings suggest that Numb may also interact with other septins such as septins 2, 9 and 10, which were also identified with a high level of confidence as Numb interacting proteins by our LC/MS/MS analysis. Our data to not allow us to determine if Numb binds directly to these septins. Septins contain highly conserved regions, and, consequently, if one such region of septin 7 interacts with Numb, then many septins would be expected to directly bind Numb through the same domain. However, because septins self-oligomerize, is possible that when Numb binds to one septin, antibodies against Numb could also pull down other septins present in the septin oligomer to which Numb is bound regardless of whether or not they are also bound by Numb. “

6. The text for Figure 5 describes analysis of Septin localization in inducible Numb/Numb-like cKO muscle, but the figure indicates only Numb is knocked out. Please clarify.

Response: We apologize for this oversight on our part. The Legend to Figure 5 has been corrected.

7. Supplementary Figure 2 seems to show that TAM treatment increases Numb expression. Please clarify. Also, please correct reference 9.

Response: The figure was incorrectly labeled. We apologize for this oversight and have corrected the figure in the revised manuscript.

**Reviewer #2 (Recommendations for The Authors):**
Overall, the manuscript is well written. I do have a few minor issues/concerns, which are detailed below.Abstract: Please be a little more specific regarding which where the tissue came from (i.e. humans, mice, cell) when referring to your previous studies.

Response: The abstract has been revised as requested.

Introduction: Please be more specific regarding the technique used for detecting ultrastructural changes. I assume it was done with TEM, but the reference is listed as an "invalid citation" in your reference list.

Response: The introduction was revised as requested and the citation was updated to reference a valid citation.

Methods / Numb Co-Immunoprecipitation: Please indicated the level of confluency of the C2C12 cells as this will alter gene expression.

Response: As indicated in the updated Methods section, confluent C2C12 cells were switched to differentiation media (low serum) for seven days. When harvested, the cells had differentiated and fused into myotubes.

Methods / Immunohistochemical Staining: The first sentence needs to be edited regarding plurality and grammar.

Response: Thank you for this comment. The text was revised accordingly.

Results / GWAS and WGS Identify...: Please spell out phosphodiesterase (I assume) for PDE4D

Response: This change was incorporated in the text.